# 3D-Printed Microneedles with Controlled Structures for Drug Delivery Study in an Ex Vivo Model

**DOI:** 10.3390/mi16111249

**Published:** 2025-10-31

**Authors:** Dong Wei, Weixiong Yang, Xiang Song, Fu Liu

**Affiliations:** 1Department of Oncology, Second Hospital of Shanxi Medical University, No. 382 Wuyi Road, Taiyuan 030001, China; wderyuan@163.com; 2School of Mechanical Engineering & Automation, Beihang University, No. 37 Xueyuan Road, Beijing 100191, China; 3School of Aerospace Engineering, North University of China, No. 3 Xueyuan Road, Taiyuan 030051, China

**Keywords:** 3D printing, microneedle arrays, controlled structures, drug delivery

## Abstract

MN arrays are highly beneficial for transdermal drug delivery, primarily due to reduced pain and improved compliance. However, complex processing and restricted design freedom limit traditional fabrication. We utilized stereolithography (SLA) 3D printing as a breakthrough method to achieve the one-step production of MN arrays with customized geometries (structure, size, tip angle), enabling tunable release profiles and eliminating cumbersome manufacturing steps. After fabricating conical and grooved MN arrays and validating them in an ex vivo porcine skin model, we confirmed their functionality. Notably, the grooved design provided excellent penetration, requiring only 2.2 N of force for full insertion (at 18.92° tip angle), and demonstrated an enhanced drug loading capacity of 4.8 μg released over 2 h. These results underscore the significant potential of 3D-printed, channel-structured MN arrays as a new generation of high-performance transdermal delivery devices.

## 1. Introduction

Microneedle arrays are sharp, microsized needles that promote transdermal drug delivery by physically penetrating the outer layer of the skin [1]. Compared to traditional needle-based injections, microneedles exhibit multiple advantages, including efficiency in drug adsorption and delivery, minimally invasive injections, the alleviation of injection pain to improve patient compliance and safety [2,3], and a high loading capacity and dose sparing [4]. Furthermore, for non-skilled use, microneedle-based injection exhibits simple application and ease of use, with the potential for self-administration [5]. Thus, microneedle technologies have been explored as a promising approach for transdermal delivery, highlighting their utility in administering therapeutic agents (such as alendronate, heparins, insulin, and vaccine components) and functional molecules (such as RNA and DNA) [6,7,8,9].

According to the different types of application requirements, microneedles are generally categorized into four types, namely solid, hollow, coated, and dissolvable microneedles [10]. Ultimately, the selection of these strategies is determined by the application’s design constraints, which encompass dosing needs, kinetics, and stability. To date, a broad range of production methods have been employed to fabricate needles with microscale structures, such as lithography and etching, electroplating, laser cutting, and micromolding [11,12]. However, existing microneedle fabrication techniques largely depend on indirect micromolding. Because these methods necessitate a multitude of time-consuming and cumbersome steps, they ultimately lengthen the time needed to develop new designs and result in resource waste. Although numerous studies have demonstrated successful applications [13,14,15], these types of methods have technical limitations that prevent certain types of microneedles from being produced, such as porous, large, or channel structures.

To address the limitations of traditional approaches, 3D printing techniques have been proposed to manufacture microneedles with controlled features by one-step processing [16]. With this flexible manufacturing method, various types of MN structures have been designed and printed to systematically explore the design parameters for optimized MN structures. Depending on the controllable processing characteristics, conical microneedles and customized microneedles with complex structures can be successfully printed. Among the various structures, microneedles with channel features can administer large compounds in clinically useful amounts if required [17]. In comparison, hollow microneedles can be utilized to deliver high-molecular-weight compounds such as antigens, oligonucleotides, or particulate formulations as solutions or suspensions into the skin [18].

With 3D printing techniques, microneedles have been produced for various applications in the field of transdermal drug delivery. For example, using fused deposition modeling (FDM) technology, researchers have developed post-chemical etching fabrication to obtain sharp-tipped microneedles with blunt columnar structures [19]. They further demonstrated how the microneedles could be exploited to load small-molecule drugs and release them. FDM printing is advantageous due to its simplicity and low cost, but it is limited by its relatively low accuracy. Generally, high-precision and complex microneedle arrays can be printed through digital light projection (DLP) and two-photon polymerization (TPP), both of which employ light as the primary energy for fabrication, but they are not widely adopted. For example, personalized microneedle arrays were designed and printed by digital light projection on a contoured surface to treat trigger finger. Here, microneedles were designed to match the outline of the hand and effectively increased anti-inflammatory drug delivery across the wavy skin surface of the affected fingers [20]. The above two methods rely on precise instrumentation and are therefore more costly. Stereolithography (SLA) is another method that uses light to cure polymers to print MNs with high precision, an affordable cost, and biocompatibility. In comparison, SLA is widely adopted with regard to its many advantages, such as the flexibility of design, relatively high resolution (some with features as fine as 30 µm), simple operation, rapid fabrication, and low cost. For example, researchers have utilized this technology’s pyramid features for the skin-based drug delivery of insulin. The results showed that the obtained microneedles were successfully inserted into the skin, and insulin was released rapidly within 30 min [21]. Furthermore, hollow microneedles were also successfully fabricated using SLA for the effective delivery of ceftriaxone sodium. The results indicate that this fabricated HMN system offers a practical solution for the transdermal delivery of a range of drugs characterized by a high molecular weight and gastrointestinal instability [22]. However, the microneedle structures designed in the aforementioned studies were relatively simplistic, and the precision of the fabricated microneedles remains insufficient.

The diversity in microneedle design, including the height [23], tip angles (i.e., aspect ratios, aspect ratio = height/width) [24], density of microneedle arrays, and microneedle shape [25], has been verified to influence their ability to enhance transdermal permeation and control the drug loading capacity. Recently, the need for improved control over microneedle designs has been highlighted. Many works have been conducted to identify appropriate parameters for microneedle fabrication and optimal delivery. For example, cone- and pyramid-shaped microneedles were designed by 3D printing to compare the pierce force and insulin release. The results showed that conical microneedles required the least force to penetrate porcine skin, and similar release profiles were obtained. In another study, researchers fabricated microneedles with microfluidic channels using 3D printing. Then, microneedles coated with fluorescein were tested to demonstrate delivery into the skin of excised rabbit ears [26]. Additionally, bioinspired needles with barbed structures were created by 3D printing to explore the size scale effect on the insertion force and further reduce it. Subsequently, an ex vivo insertion test was conducted, showing that honeybee-inspired needles decreased the insertion force (by 21–26%) regardless of the needle scale [27]. While these studies have confirmed the efficacy of different needle structures in the delivery system, the research scope is limited, with few investigations into how the needle body geometry influences both drug loading and the insertion force.

Based on the above, here, microneedles with complex channel structures were designed, and their performance during the drug delivery process was compared. Cone-shaped microneedles with varying sizes, tip angles, and curvatures were fabricated using a conventional SLA printer, and their ability to penetrate porcine skin was successfully verified in an ex vivo model. This method demonstrated flexibility in fabricating microneedles with tunable features. Specifically, three types of microneedles with controlled structures were prepared for drug release, namely spiral grooves, cambered grooves, and hollow slots. In our proposed design, the grooved microneedles demonstrated a significant reduction in the insertion force, as well as the capacity to load and deliver large quantities of drug. Finally, using an ex vivo model, we evaluated the drug release of coated microneedles, demonstrating the feasibility of this technique to deliver therapeutic components to the derma.

## 2. Materials and Methods

### 2.1. Materials

The printing resin was composed of methacrylate, acrylate-based oligomers/monomers, photoinitiators, and pigments and additives. Poly(methyl methacrylate) was purchased from Formlabs Inc. (Somerville, MA, USA). Sodium alginate (CP, viscosity: 100–300 cp) and dextran (C_6n_H_10n_O_5n_ MW 7000) were purchased from Aladdin Chemistry Co., Ltd. (Shanghai, China). Methylene blue (MW 373.9) was from Sigma Aldrich Co., Ltd. (Shanghai, China). All materials were used as received.

### 2.2. Microneedle Fabrication

Arrays of various needles, i.e., conical, grooved, and hollow microneedles, were produced using a Form2 stereolithography (SLA) printer by Formlabs Inc. (Somerville, MA, USA), with a wavelength of 405 nm and laser power of 250 mW. The printer featured a laser spot size of 140 µm, a layer thickness of 25 µm, and a short exposure time of approximately 5–8 s per layer. The fabrication process was initiated by generating stereolithography (STL) files using the Inventor (Autodesk^®^ Inc., San Francisco, CA, USA) computer-aided design (CAD) software. These files were subsequently exported to the 3D Preform Software (Formlabs Inc., Somerville, MA, USA) for pre-processing. During the printing stage, ultraviolet (UV) light was precisely projected through the transparent window at the base of the printer, enabling the selective photopolymerization (curing) of each cross-sectional layer. The three-dimensional construct was built incrementally, with each successive layer being cured onto the preceding layer via bottom-up irradiation. A summarized overview of this fabrication process is provided in Figure 1. To improve the mechanical properties, all printed microneedles were washed in isopropyl alcohol to remove excess resin and then cured under ultraviolet radiation with a wavelength of 405 nm for 20 min. To observe the structure, the as-obtained microneedles were examined by scanning electron microscopy (SEM) with an accelerating voltage of 20 kV, and some of them were also imaged using an optical microscope.

#### 2.2.1. Microneedle with Simple Structure

To demonstrate the ability to adjust the microneedle size, CAD files of conical microneedles were created using Inventor 2015. These files were then sliced into layers with a thickness of 25 μm via the preprocessing software PreForm. Subsequently, the printer fabricated microneedles across a wide range of scales, with the smallest one having a diameter of 0.25 mm and a height of 0.65 mm. Considering that the skin surface is non-planar and that specific joint regions of the human body are susceptible to damage, the fabrication of microneedle arrays on curved substrates is essential to meet practical demands. Furthermore, a large-scale 8 × 8 microneedle array was developed to enable broad-area therapeutic applications.

#### 2.2.2. Microneedle with Channel Structures

To explore the effects of the structure on the insertion force and coating release, microneedles with hollow and grooved features were generated. The channel structures fabricated on the microneedle bodies were capable of minimizing the contact area between the microneedle and the target tissue during the insertion process. To guarantee that penetration occurred via the needle tip, the delivery port for the hollow microneedles was positioned non-coaxially relative to the needle’s axis, with the relative offset distance being variably settable. The printer was also used to fabricate grooved microneedles possessing controllable cross-sectional parameters, such as spiral and cambered structures. Crucially, these complex geometries enabled the delivery of a greater payload of drug substances when benchmarked against solid microneedle designs.

### 2.3. Mechanical Testing of Microneedle Arrays in Ex Vivo Model

The mechanical efficacy of the microneedles, which is pertinent for transdermal drug delivery, was quantitatively determined through an ex vivo penetration assay. The model utilized porcine skin, commercially acquired, based on its established utility as a reliable surrogate for human skin [27]. Given that microneedles’ geometric parameters have been proven to directly modulate the required piercing force, the current investigation employed microneedles differentiated by their tip angles and structural designs to elucidate these effects. For sample preparation, skin specimens cryopreserved at 20 °C were subjected to a 30 min period of ambient thawing, followed by precise cutting into sections exhibiting thickness uniformity. All ensuing mechanical characterization was executed on an MTS tester (Model E44, MTS Co., Eden Prairie, MN, USA), and the force–displacement data were generated by the MTS tester (E44), which used its internal load cell and displacement encoder to record the real-time force and crosshead travel under a constant insertion speed of 1 mm/min. Critically, at least three replicate tests were conducted for the samples corresponding to each parameter.

### 2.4. Cargo Release from Microneedles in Ex Vivo Model

To promote drug adhesion on the microneedles, sodium alginate was applied to increase the solution viscosity. Moreover, methylene blue was utilized as a fluorescent drug surrogate to trace the delivery of cargo. In order to prepare the solution for microneedle coating, dextran was dissolved in deionized water at 1.5% (*w*/*v*) and stirred at 40 °C for 10 min. Next, 2% (*w*/*v*) sodium alginate was added, and the mixture was stirred at 50 °C with a speed of 300 rpm for 5 h. Then, 0.05% (*w*/*v*) methylene blue was added into the mixture, and the solution was then homogenized by stirring for 30 min. The final suspension was stored at 4 °C until testing occurred.

Several methods, such as dip coating, gas jet drying, spray coating, and ink printing, have been proposed and applied for the coating of microneedles. However, currently, dip coating is the most widely applied technology [28]. To facilitate drug loading, microneedles were treated with plasma to clean and activate the surface. While drug loading was simple to prepare, the release rates were difficult to control and calculate quantitatively. To load the coating, microneedle tips were dipped into a solution and left for 5 min. The capacity for drug loading among the different structures was compared. Under the same coating time, the arrays were soaked in deionized water to dissolve the coating solution. Then, the content of methylene blue in the solution was measured and analyzed using a spectrophotometer.

The study of dye release from tip-loaded microneedles was performed in porcine skin [29]. Notably, each experimental group included over three replicates. The coated microneedle arrays were applied to the porcine skin with 10 s of thumb pressure to the array substrate and allowed to remain for 3 min. Then, we removed the porcine skin 10, 20, 30, 60, 120, and 180 min after initial insertion under a temperature of 37 °C. To assess the diffusion rate of pigment from the microneedles into porcine skin tissue, each skin specimen was first refrigerated and then cut from the center of the insertion site. An optical microscope was employed to observe and record cross-sections of the samples. Finally, the variation in diffusion with time was estimated by processing images collected via ImageJ, and the release rate was calculated by the following equation:(1)$$Release\;rate(t)=\frac{S_{t}-S_{0}}{S_{max}-S_{0}}$$
where *S*_0_ was the initial cross-section area of the porcine sample that had been dyed, *S_max_* was the final cross-section area of the porcine sample that had been dyed when diffusion became stable over time, and *S_t_* was the cross-section area of the porcine sample that had been dyed at time (*t*).

## 3. Results

Microneedles featuring diverse structures have garnered significant attention because of their potential to control the penetration depth and modulate the available volume for drug loading. With the feasibility of 3D printing techniques, microneedle arrays with diverse structures and scales have been successfully designed and printed through stereolithography (SLA) printers. By virtue of its flexible designs and high production efficiency, this method has become an attractive process for microneedle fabrication. To investigate the properties of the microneedles, penetration and drug release studies were performed using ex vivo models.

### 3.1. Microneedle Arrays Fabricated via Stereolithography Printing

Microneedles with diverse scales and structures were printed accordingly. The process of altering the microneedle types only required the generation of a homologous CAD file. Figure 2a1–a7 demonstrates the ability to adjust the microneedle scale, ranging from 0.25 mm to 1.5 mm in diameter and 0.65 mm to 3 mm in height, all while maintaining a desirable microneedle morphology. Remarkably, the smallest cone-shaped microneedle that we printed was 0.65 mm in height and 0.25 mm in diameter. Compared with traditional molding processing, which requires days to produce the necessary templates, an 8 × 8 microneedle array on a 1 cm^2^ base here only took 55 min. Similarly, microneedles with tip angles varying from 18.92° to 45.24° were readily generated, as shown in Figure 2b1–b8. As shown in Figure 2c1–c4,d1–d3, this method successfully achieves highly consistent, large-area microneedle arrays and enables the printing of microneedle structures onto non-planar substrates. Although simple, solid microneedles with cone geometries have historically been the mainstay of the field, more complex structures offer distinct advantages in both penetration and drug loading efficiency. By 3D printing, defined microneedles with three channel features were fabricated, as shown in Figure 3. Figure 3a_1_–h_1_ first display the spiral groove microneedles with different screw pitches and revolutions. The design of spiral grooves may enable flow resistance among solutions or suspensions to retain the drug loading at the maximum capacity. Microneedles with cambered groove structures are shown in Figure 3a_2_–h_2_. Three types of hollow microneedles were also fabricated, as shown in Figure 3a_3_–h_3_. Defined and designed based on their sectional parameters, these structures are beneficial in storing or delivering a broad range of therapeutic compounds. This flexible fabrication approach, therefore, is well suited to meet the diverse needs of different disease types and severities.

### 3.2. Mechanical Testing of Microneedle Arrays in Ex Vivo Model

The geometry of the needle body is a critical factor influencing needle insertion. For the penetration assessment, microneedles were fabricated encompassing seven tip angles and three types of channel structures. Each microneedle successfully pierced the skin, and no structural fracture was observed. The resulting force–displacement curves recorded during the penetration event are presented in Figure 4. Figure 4a illustrates that a smaller tip angle corresponds to the minimum force generated during skin insertion. This relationship shows that the microneedle sharpness inversely affects the required insertion force. The minimum force recorded is approximately ~2.2 N (18.92° tip angle at 0.85 mm depth), which favorably compares to and is lower than the forces reported for metallic microneedles [30]. Otherwise, the maximum force measured is nearly ~6.6 N, with a 200% improvement, being far lower than the load range of the skin (0–20 N) [31]. We also measure the loads of the defined structures. As shown in Figure 4c, microneedles with channel structures puncture the skin model with less pressure than solid ones. Researchers have found that the force of insertion may correlate with pain [32], which suggests that grooved microneedles can alleviate pain with minimal invasiveness. We observed moderate differences in the penetration force among the three microneedle types. Specifically, the spiral groove array microneedles generated a larger insertion force than the other channel microneedles. These variations are attributed to differences in the contact area between the microneedles and the skin. Owing to the inherent elastic properties of skin, the presence of channel structures generally reduces the contact area during insertion, resulting in a slight decrease in the overall insertion force. As seen in Figure 4e, it is clear that the contact area between the microneedle and the skin is decreased when the arrays are inserted into the elastic skin, resulting in a decrease in piercing pressure. In addition, according to Figure 4d, the penetration force gradually decreased while the porcine skin was under cyclic loading. This indicates that the porcine skin was successfully punctured.

### 3.3. Cargo Release from Microneedles in Ex Vivo Model

Figure 5a compares the drug loading capacities of the conical and grooved microneedles. Microneedles incorporating the designed structures show a significant increase in loading capacity, improving by 108%—from 2.3 μg for conical structures to ~4.8 μg for hollow structures. Among all tested designs, the spiral groove microneedles exhibit the optimal drug loading, which is almost two times higher than that achieved by the cone arrays. We conjecture that this enhancement is due to the increased surface area afforded by the channel features, which facilitates the adhesion of more coating material. Specifically, the spiral structure not only extends the relative surface area but also impedes the flow of the coated loading, thereby improving retention. The microneedle arrays featuring cambered groove and hollow structures show similar results, likely due to their analogous structural features. Taking the spiral groove as an example, the structure leads to creating concaves on the surfaces of the microneedles. Additionally, the flexible design allows operators to obtain a larger amount of loading on demand.

Figure 5b compares the drug release kinetics of the microneedles over time. The release profile for the spiral groove structures exhibits a trend similar to that of the conical design. Notably, approximately 35% of the pigment was released within the first 30 min, with the majority of the substance being released after 2 h. This result indicates the successful delivery of the drug through the pores generated in the porcine skin upon microneedle piercing. As seen in Figure 6, soon after insertion, the pigment presented at the position of penetration and started to diffuse as time progressed. It also could be found that the cross-sectional area of the microneedles with a groove structure was larger than that seen in conical ones. As illustrated in Figure 6, pigment is present at the site of penetration immediately after insertion and begins to diffuse as time progresses. Furthermore, it is observed that the cross-sectional area of the microneedles featuring a groove structure is larger than that in the conical designs. This phenomenon demonstrates that grooved microneedles can load a large amount of cargo. Together, these results indicate that the microneedle arrays that were designed and fabricated by stereolithography printing are capable of the loading of cargo and its effective release.

## 4. Discussion

Microneedles offer several advantages for drug delivery, including the alleviation of injection-associated pain and anxiety, a reduced risk of infection, and simplified operation. Consequently, microneedle-based skin injection has been successfully applied to deliver various compounds, demonstrating high effectiveness within the delivery system. However, the limitations of traditional fabrication techniques have historically constrained further development and innovation in this field. To advance the application of microneedles, 3D printing has been adopted, enabling the manufacturing of microneedles with flexible designs and customized structures via a one-step forming process. Compared to conventional micromolding techniques, 3D printing offers significant advantages in terms of time and energy efficiency. For instance, fabricating an 8 × 8 microneedle on a 1 cm^2^ base required only 55 min using this 3D printing method. This approach facilitates the rapid, high-throughput fabrication of microneedles with various parameters, including size, shape, and internal structure. As for the material cost and resin biocompatibility, this printer has been widely reported in microneedle fabrication for biological applications, featuring a low cost and excellent biocompatibility [21]. We anticipate that the benefits afforded by 3D printing will accelerate clinical research. For example, this technique allows for the clear evaluation of the effects of microneedle designs (such as shape and structure) on skin penetration. Furthermore, microneedle arrays featuring channel structures designed to control drug loading can also be easily fabricated using 3D printing.

However, the translation of 3D-printed microneedle arrays to clinical applications requires further investigation. Specifically, the biocompatibility of the microneedle device and the stability of the printing material must be seriously considered. Careful evaluation of the safety profiles of these devices is essential. Furthermore, ensuring a high degree of reaction completion during the photopolymerization process or the complete removal of residual polymer is critical for future device development and successful application. Regarding cargo loading, methods for both coating and encapsulating compounds still warrant further exploration. In addition, while this advanced method allows microneedles to be directly used for therapeutic applications, a considerable opportunity exists to combine 3D printing technologies with existing micromolding techniques. The feasibility and efficiency of 3D printing make it an indispensable technique for generating microneedle master templates. These masters can then be replicated using polymer or polydimethylsiloxane (PDMS) molds to efficiently fabricate microneedles from a wider variety of materials.

The specific advantage of the stereolithography (SLA) technique demonstrated herein is its ability to rapidly and flexibly alter microneedle designs. This capability allows for the fabrication of microneedles encompassing diverse scales and tip angles. Furthermore, this method proves valuable for producing microneedle arrays even on non-planar surfaces. Simultaneously, to enhance the drug loading capacity of the devices, defined microneedles incorporating channel features have also been successfully fabricated. To assess their penetration, we conducted mechanical tests using ex vivo porcine skin. The results demonstrate that microneedles incorporating channel structures require less force for skin penetration than their cone-shaped counterparts. The possible mechanism behind this is that the spiral and cambered groove structures can reduce the puncture contact area and frictional resistance. This significantly lowers the insertion force while ensuring effective stratum corneum penetration and biocompatibility. Similarly, spiral grooves, with a larger specific surface area and longer diffusion path, enhance the drug loading capacity and extend the sustained-release duration. Cambered grooves, via smooth and short channels, achieve rapid initial release followed by continuous drug delivery, meeting diverse administration requirements. The reduced insertion force associated with grooved microneedles suggests that they may cause less patient pain. Minimizing pain during needle insertion is crucial, as discomfort is a primary contributor to patient distress, poor adherence, and, in extreme cases, the development of needle phobia and anxiety. Therefore, mitigating pain in human subjects is crucial in advancing microneedle development. To investigate the loading capacity, we conducted a drug release study where the drug loading of microneedles with different structures was assessed. The coating solution was prepared using sodium alginate to increase the viscosity, with methylene blue serving as a common index for diffusion measurement. Microneedles incorporating channel structures successfully increased the drug loading capacity, thereby demonstrating their potential for large-volume drug delivery.

Drug release is achieved into the dermal microcirculation through the micropores created by the needle tips. In summary, the channel-structured microneedles developed in this study demonstrate the capacity for massive and quantitative drug loading and delivery. The results from the ex vivo penetration and drug release tests confirm that the printed microneedles could successfully penetrate porcine skin and achieve therapeutic delivery. Moreover, the designed channel features facilitate easier insertion into the skin, underscoring their practicality for future drug delivery applications. While this study validated the feasibility of 3D-printed microneedles, it lacked an in-depth analysis of how structural parameters affect drug delivery performance. Future works will focus on defining the relationships between these structural parameters and the resultant mechanical properties and drug delivery efficiency, supplemented by a qualitative biocompatibility evaluation.

## 5. Conclusions

In this work, a 3D printing technique was introduced to fabricate microneedles for transdermal delivery. Various multiscale and designed structures were printed through a stereolithographic printer. The penetration and release tests showed that the microneedles were successfully inserted into porcine skin and achieved drug delivery. Notably, three types of channel-based microneedles were designed and printed, and they have demonstrated the ability to reduce the required insertion forces, while improving cargo loading and enabling certain release. Spiral and cambered groove structures have been demonstrated to load and deliver components. In summary, this flexible, convenient, personalized, and rapid approach might be preferred for the fabrication of microneedles in transdermal drug delivery systems in the future.

## Figures and Tables

**Figure 1 micromachines-16-01249-f001:**
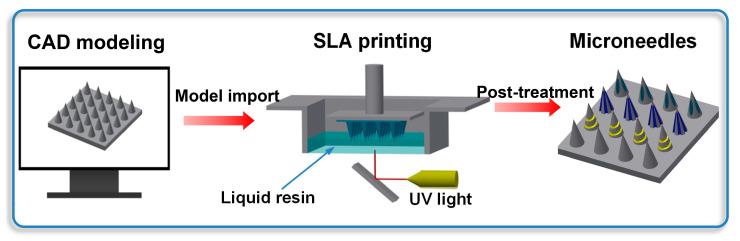
Schematic illustration of microneedle production.

**Figure 2 micromachines-16-01249-f002:**
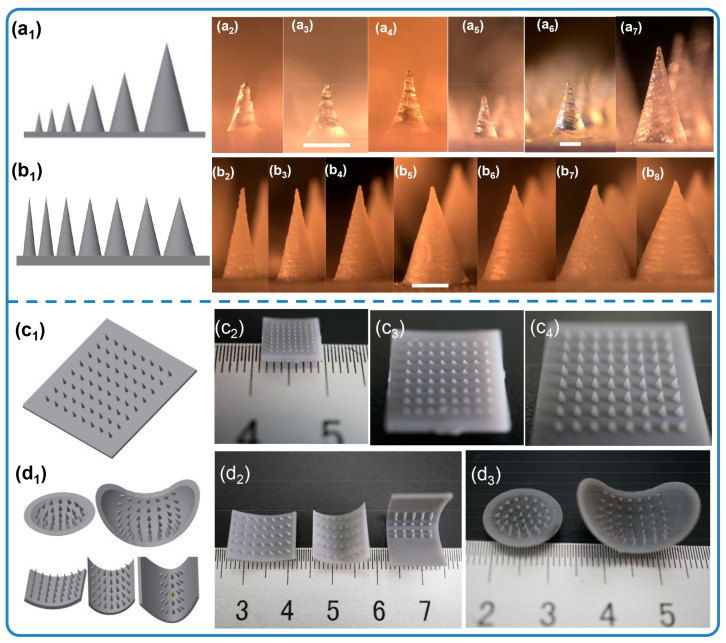
Solid microneedles with different scales (**a_1_**–**a_7_**), (**a_1_**) 3D model, (**a_2_**–**a_7_**) optical images of microneedles: (**a_2_**) 0.25 × 0.65 mm, (**a_3_**) 0.3 × 0.8 mm, (**a_4_**) 0.5 × 1.2 mm, (**a_5_**) 0.8 × 1.6 mm, (**a_6_**) 1 × 2 mm, (**a_7_**) 1.5 × 3 mm (Φ × h) (scale bar = 500 µm). Solid microneedles with different tip angles (**b_1_**–**b_8_**), (**b_1_**) 3D model, (**b_2_**–**b_8_**) optical images of MNs: (**b_2_**) 1 × 3 mm, (**b_3_**) 1.2 × 3 mm, (**b_4_**) 1.5 × 3 mm, (**b_5_**) 1.8 × 3 mm, (**b_6_**) 2 × 3 mm, (**b_7_**) 2.2 × 3 mm, (**b_8_**) 2.5 × 3 mm (Φ × h) (scale bar = 1 mm). Optical images of 3D-printed MNs, (**c_1_**) 3D model of large arrays, (**c_2_**,**c_3_**) 0.25 × 0.65 mm, (**c_4_**) 0.8 × 1.6 mm; (**d_1_**) 3D model of non-plane arrays; (**d_2_**) surface structure arrays, 0.8 × 1.6 mm; (**d_3_**) sphere structure arrays, 0.8 × 1.6 mm (Φ × h).

**Figure 3 micromachines-16-01249-f003:**
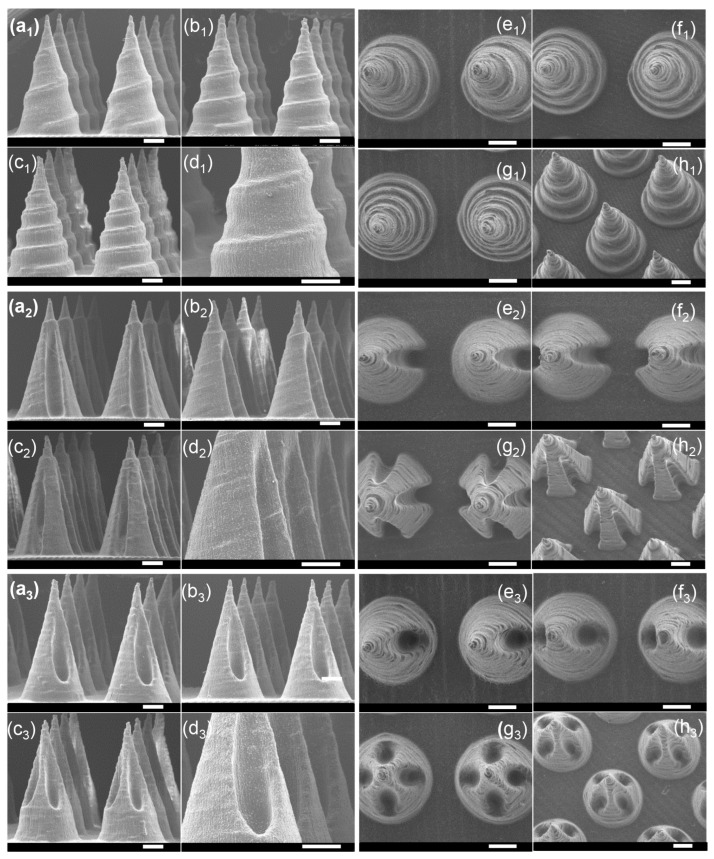
SEM images of microneedles with groove structures. (**a_1_**–**h_1_**) Three types of spiral groove microneedles, (**a_1_**–**d_1_**) front view, (**e_1_**–**h_1_**) top view; (**a_2_**–**h_2_**) three types of cambered groove microneedles, (**a_2_**–**d_2_**) front view, (**e_2_**–**h_2_**) top view; (**a_3_**–**h_3_**) three types of hollow hole microneedles, (**a_3_**–**d_3_**) front view, (**e_3_**–**h_3_**) top view (scale bar = 500 μm).

**Figure 4 micromachines-16-01249-f004:**
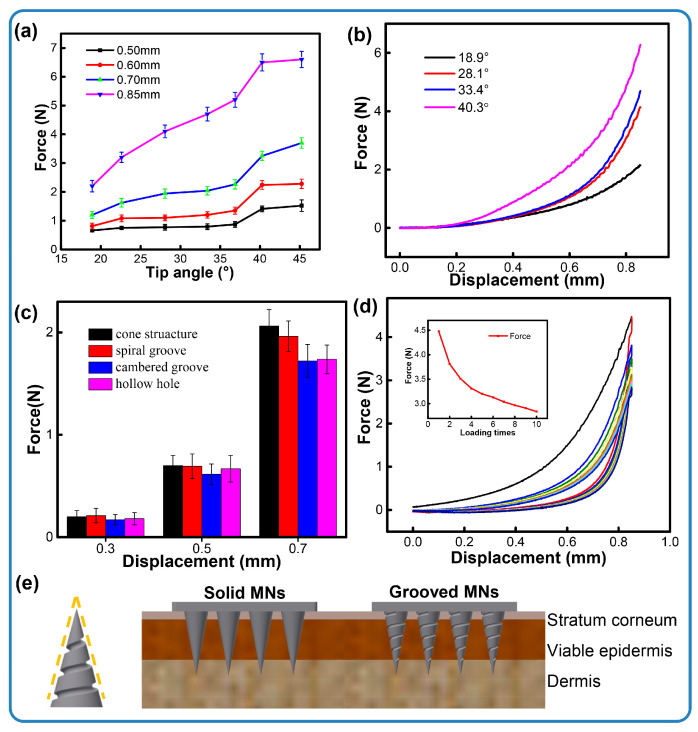
Insertion forces of microneedles. (**a**) Different tip angles within insertion depths, (**b**) the force process under different tip angles, (**c**) channel structures within insertion depths, (**d**) cycle loading of conical microneedles, (**e**) the contours of spiral groove microneedles and a schematic illustration of the penetration of the skin with different types of microneedles.

**Figure 5 micromachines-16-01249-f005:**
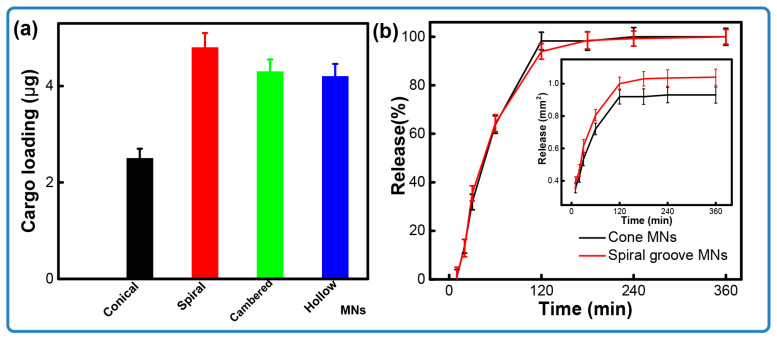
The release of coated microneedles. (**a**) Comparison of loading between solid and groove microneedles; (**b**) release of microneedles over time.

**Figure 6 micromachines-16-01249-f006:**
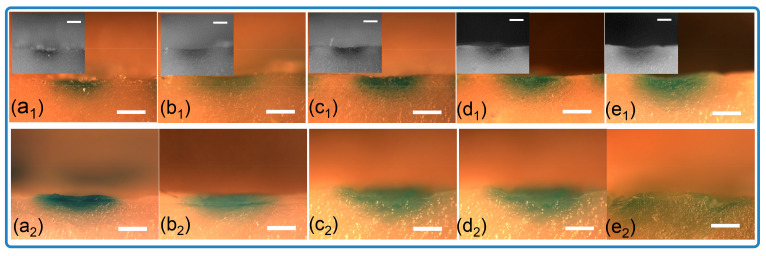
Optical images of cross-sections for porcine samples inserted with loaded microneedles. (**a_1_**–**e_1_**) Solid microneedles, (**a_1_**) 10 min, (**b_1_**) 20 min, (**c_1_**) 30 min, (**d_1_**) 1 h, (**e_1_**) 2 h; (**a_2_**–**e_2_**) grooved microneedles, (**a_2_**) 10 min, (**b_2_**) 20 min, (**c_2_**) 30 min, (**d_2_**) 1 h, (**e_2_**) 2 h (scale bar = 1 mm).

## Data Availability

The original contributions presented in this study are included in the article. Further inquiries can be directed to the corresponding authors.

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
