# Peer review of "3D-Printed Microneedles with Controlled Structures for Drug Delivery Study in an Ex Vivo Model"

_micromachines, 2025, doi:10.3390/mi16111249_

Round 1
Reviewer 1 Report
Comments and Suggestions for Authors
The manuscript demonstrates strong experimental execution using stereolithography for microneedle fabrication, but has several weaknesses that limit its clarity and scientific rigor. The article provides meaningful experimental results on microneedle design, mechanical testing, and ex vivo release. The novelty lies in fabricating different channeled structures using 3D printing to achieve controlled release. However, the manuscript suffers from linguistic errors, insufficient methodological detail, and weak data discussion. The overall presentation requires substantial improvement before publication.
Extensive grammatical and stylistic errors (e.g., “the grooved microneedle we designed showed the penetration ability…”). The text often uses incorrect tense, non-native phrasing, and inconsistent terminology such as “cargo loading” and “microneedles were arrays.”
Lack of quantitative and statistical analysis: Results (e.g., penetration force, release rate) are described qualitatively or with single values without replicates, error bars, or statistical significance.
Figures referenced (e.g., Figures 2–6) are not clearly described in captions; data presentation lacks scale consistency and quantitative clarity.
Important experimental parameters (number of samples, replicates, error analysis, calibration of instruments) are missing, limiting reproducibility.
The Discussion section restates results without integrating mechanistic explanation or comparison with literature beyond superficial referencing.
Typographical inconsistencies (spacing, punctuation, missing units such as “°C” and “μm”).
Redundant use of terms (“grooved microneedles were fabricated by the printer” repeated multiple times).
The abstract is wordy and should state the hypothesis, method, main numerical findings, and significance clearly.
The study has potential but requires substantial English editing, enhanced data analysis, and improved figure quality. Statistical treatment and deeper comparison with published microneedle studies are essential to strengthen scientific credibility.
Comments on the Quality of English LanguageExtensive grammatical and stylistic errors (e.g., “the grooved microneedle we designed showed the penetration ability…”). The text often uses incorrect tense, non-native phrasing, and inconsistent terminology, such as “cargo loading” and “microneedles were arrays.”
Reviewer 2 Report
Comments and Suggestions for Authors
Reviewer Comments to the Authors
Date: October 20, 2025Manuscript Title: 3D Printed Microneedles with Controlled Release for Drug Delivery Study in an Ex-Vivo Model
Recommendation: Major Revision. If not addressed, rejection.
The manuscript reports the fabrication of stereolithography (SLA)-printed microneedles with various geometries and evaluates their performance in an ex vivo porcine skin model. While the approach is timely and relevant, the manuscript lacks sufficient methodological detail, clarity in claims, and critical context regarding novelty and translational value. Major revisions are required to clarify the contribution and strengthen the scientific rigor.
Comments:
-
The manuscript does not clearly establish the novelty of the work compared to existing SLA-printed grooved or channel microneedles. Authors should explain how their designs offer specific advantages and cite relevant recent studies.
-
Claims of “controlled release” are not supported—release profiles are nearly identical across designs. Revise the language or provide quantitative evidence demonstrating a tunable release profile based on geometry.
-
Key experimental details are missing: number of replicates, error bars, and statistical analysis for mechanical testing and drug loading/release. Include standard deviations and clarify whether observed differences are significant.
-
The mechanical testing methodology is insufficiently described. Was the entire microneedle array inserted at once? Include schematic or photographic documentation.
-
Drug loading quantification lacks detail. Explain how methylene blue was measured (e.g., calibration curve), and how coating repeatability was ensured.
-
The area-based image analysis (Equation 1) used to assess release is not validated. Clarify how it relates to drug delivery dose and cite appropriate references.
-
The manuscript does not critically evaluate the advantages of SLA microneedle fabrication over conventional methods. Authors should discuss whether the approach is scalable and commercially viable.
-
The influence of needle design parameters (e.g., groove type, tip angle) on mechanical strength and release should be addressed more explicitly. Are these features tunable for different delivery profiles?
-
Reference list and in-text citations are inconsistent, especially after reference [26]. Review and correct all numbering and align citations accurately.
-
Language, grammar, and formatting errors are frequent and distracting. Examples include “drag loading,” “costed 55min,” and “there kinds of microneedles.” A full language revision is necessary.
-
Several figure captions are unclear or contain typographical errors. Ensure consistent labeling, correct units, and complete descriptions for all figures.
-
Conclusions overstate the findings. Phrases like “widely utilized in the future” or “controlled release” are not justified by the presented data and should be revised.
Reviewer 3 Report
Comments and Suggestions for Authors
The development of microneedle arrays utilizing stereolithography (SLA) 3D printing and their assessment in an ex-vivo porcine skin model for drug release investigations are the subjects of an intriguing and topical study presented in this manuscript. The subject is pertinent to the readership of Micromachines, particularly in light of the quick development of additive manufacturing for use in biomedicine. The experimental setup is well-thought-out, and the results confirm that 3D-printed microneedles with precise geometries for transdermal administration are feasible. To improve the approach's rigor and reproducibility, a number of crucial concerns in data presentation, figure clarity, methodology description, and scientific discussion need to be addressed.
- There is not enough information in the Materials and Methods section to support replication. It is necessary to include details about the resin composition, laser power, exposure parameters, curing conditions, and sample preparation procedures.
- Provide details on the methods used to generate the force-displacement and penetration depth data. Were the skin samples hydrated, and how was the microneedle alignment preserved during insertion?
- Analyze all quantitative data (penetration force, loading capacity, and release rate) statistically (mean ± SD, n values).
- Provide quantitative comparisons of the loading efficiency and insertion force of various microneedle kinds (e.g., box plots or bar charts).
- For clarity, the schematic images (such as Figure 1) ought to be redone and labeled with the appropriate process phases.
- Only a basic description of the release kinetics is given. To comprehend the release process, the authors ought to incorporate the release profile appropriate to popular models (such as Higuchi and Korsmeyer-Peppas).
- Please supplement the qualitative conclusions from the optical pictures (Figure 6) with quantitative diffusion area or intensity analysis.
- Talk about the value of using methylene blue as a model drug and how the findings could be applied to actual therapeutic medicines.
- Rather than providing a critical analysis, the Discussion section restates the findings. The authors should interpret the effects of geometric modifications (such as tip angle and groove pattern) on the release behavior and mechanical performance.
- Comparisons of the accuracy, cost, and clinical translation compatibility of alternative fabrication methods (such as DLP, FDM, or TPP) should be included.
Round 2
Reviewer 1 Report
Comments and Suggestions for Authors
The authors extensively revised the paper in response to the referee's comments. The manuscript can now be accepted for publication.
Comments on the Quality of English LanguageExtensive grammatical and stylistic errors (e.g., “the grooved microneedle we designed showed the penetration ability…”). The text often uses incorrect tense, non-native phrasing, and inconsistent terminology, such as “cargo loading” and “microneedles were arrays.”
Author Response
Thanks for your review comments.
Reviewer 2 Report
Comments and Suggestions for Authors
Thank you for the author's revisions. The authors have addressed most of the concerns raised; however, several key points still need improvement:
- The manuscript does not clearly demonstrate how your SLA microneedles are a novel approach compared to prior grooved/channel designs. Please cite and contrast specific recent studies and highlight your unique contributions.
- The current discussion is too general. To address commercial viability, add specific comments on fabrication time per array, production scalability, material cost, and resin biocompatibility concerns.
- Add error bars and standard deviations to figures. Report statistical comparisons between groups. Include a schematic of the test setup for clarity.
- Please include the Study Limitations discussing, Ex vivo model constraints, Use of methylene blue as a drug surrogate, Unvalidated drug release quantification, Material biocompatibility, etc.
Reviewer 3 Report
Comments and Suggestions for Authors
The authors' comprehensive and helpful edits to the manuscript "3D printed microneedles with controlled release for drug delivery study in an ex-vivo model" are greatly appreciated. The updated version has been greatly enhanced, and I applaud the authors for their thorough answers and thoughtful evaluation of the reviewer's recommendations. The manuscript's presentation and content have both been significantly improved. I think the manuscript is now ready for publishing following a few minor editorial checks.
Author Response
Thanks for your review comments.